# Experimental Investigation on the Shear Behaviour of Stud-Bolt Connectors of Steel-Concrete-Steel Fibre-Reinforced Recycled Aggregates Sandwich Panels

**DOI:** 10.3390/ma14185185

**Published:** 2021-09-09

**Authors:** Arash Karimipour, Mansour Ghalehnovi, Mohammad Golmohammadi, Jorge de Brito

**Affiliations:** 1Department of Civil Engineering, University of Texas at El Paso (UTEP) and the Member of Center for Transportation Infrastructure Systems (CTIS), El Paso, TX 79968, USA; akarimipour@miners.utep.edu; 2Department of Civil Engineering, Ferdowsi University of Mashhad, Mashhad 9177948944, Iran; 3Civil Engineering Department, Faculty of Engineering, University of Torbat Heydarieh, Torbat Heydarieh 9516168595, Iran; m.golmohammadi@torbath.ac.ir; 4Department of Civil Engineering, Architecture and Georresources, Instituto Superior Técnico, Universidade de Lisboa, 1649-004 Lisbon, Portugal

**Keywords:** bolt connectors, fibre-reinforced concrete, steel-concrete-steel sandwich, shear behaviour, ultimate load

## Abstract

Steel-concrete-steel (SCS) sandwich panels are manufactured with two thin high-strength steel plates and a moderately low-density and low-strength thick concrete core. In this study, 24 specimens were produced and tested. In these specimens, a new stud-bolt connector was used to regulate its shear behaviour in sandwich panels. The bolts’ diameter, concrete core’s thickness and bolts’ spacing were the parameters under analysis. Furthermore, the concrete core was manufactured with normal-strength concrete and steel fibres concrete (SFC). Steel fibres were added at 1% by volume. In addition, the recycled coarse aggregate was used at 100% in terms of mass instead of natural coarse aggregate. Therefore, the ultimate bearing capability and slip of the sandwich panels were recorded, and the failure mode and ductility index of the specimens were evaluated. A new formula was also established to determine the shear strength of SCS panels with this kind of connectors. According to this study, increasing the diameter of the stud-bolts or using SFC in sandwich panels improve their shear strength and ductility ratio.

## 1. Introduction

Nowadays, waste construction materials have become a major challenge in the field of environmental pollution. As per prior reports, between 2012 and 2014, nearly 370 million tonnes of waste construction materials were dumped in nature, which accounts for approximately 70% of the total waste construction materials generated in each year [1]. Therefore, if the production of construction waste maintains this tendency, 430 million tonnes of waste will be dumped annually. In addition, some natural phenomena such as tornadoes, earthquakes, and floods play an effective role in increasing construction wastes [2,3,4,5]. While the recycling of old building rubble has become very popular, much of it is still dumped in nature. Velay-Lizancos et al. [6] declared that reusing waste materials to manufacture new concrete can play a crucial role in keeping environment safe and clean. Furthermore, the use of coarse recycled aggregates (CRA) as a substitute of coarse natural aggregates (CNA) substantially decreases CO_2_ production by about 15–20% [7]. As a result, the importance of using recycled materials and the increasing use of CRA concrete is clear, and understandable. Many investigations have been performed on using of CRA to produce conventional recycled aggregate concrete. Thus, so far, a large number of disadvantages of CRA use on the general behaviour of a concrete mixes have been detected, largely because old mortar is adhered to the CRA surfaces [8,9,10]. Hansen and Narud [11] showed that, based on the aggregates size, 25% to 60% of the mortar can attach to the aggregates surface. Additionally, with an increase in the RCA content, the mechanical properties of concrete decreased [12,13,14]. Amer et al. [15] found the same observations and declared that raising RCA substitution content led to reducing the compressive strength of concrete. On the other hand, some research indicated an enhancement effect of CRA on the structural behaviour of concrete beams due to their larger broken surface, relative to that of CNA. Therefore, the behaviour of concrete members still needs evaluation.

On the other hand, prior investigations indicate that using fibres in concrete mixes substantially enhances the mechanical and rheological characteristics of concrete. Fibres decrease cracks width as well as their expansion [16,17]. In addition, the bridging role of fibres interacts with the paste at the level of micro-cracks and postpones cracks propagation as well as improving the resistance of concrete when enough fibres content is used [18]. Therefore, when tensile strength of concrete increases, micro-cracks accumulate and convert into large cracks, and fibres prevent them from opening and propagating by efficiently bridging them. This post-peak macro-crack bridging is the main feature improving the properties of fibre-reinforced concrete (FRC). Using a low to intermediate fibres content does not improve the tensile and flexural strengths of concrete and only slightly enhances the energy absorption and durability in the post-cracking state. Conversely, the addition of high fibres fractions improves the tensile strength, strain-hardening performance before localization and toughness beyond crack localization [19,20,21,22,23]. Therefore, once FRC is conducted under a flexural load, fibres enhance the bending behaviour. Additionally, the use of fibres at the cracking location leads to delaying in crack propagation, reducing the decrease caused by high CRA substitution of CNA on the structural behaviour of concrete [24,25]. In another research, Niu et al. [26] evaluated the structural behaviour of FRC in water and salt freezing conditions. Experiments showed that the use of SF can substantially enhance the micro structure and tensile strength of samples. Olivito and Zuccarello [27] investigated the influence of SF on the mechanical characteristics of concrete and its classification considering SF fraction and mix-design variations. They found that the SF contents and their geometric characteristics are vital features and crack width and tensile strength of concrete went up by raising the length of SF. Köksal et al. [28] assessed the effect of water/cement ratio, tensile strength and volume of SF on the properties of concrete and provided solutions for FSC design based on the maximum fracture energy. They also found that the tensile strength of SF and the water/cement ratio directly affect the structural behaviour of FRC.

On the other hand, steel-concrete-steel (SCS) panels comprise two outer steel plates and a concrete core. An interrelated material or connectors are generally used to guarantee an adequate connection between the concrete core and steel plates. Besides, using connectors improves the ductility and the biaxial performance of SCS members [29,30]. In recent years, SCS sandwich plates have been widely used to manufacture oil and gas silos in the Arctic region and shear walls for offshore structures. This structure can withstand the high pressures of ice up to 45 MPa [29]. The different kinds of connectors that have been proposed for SCS sandwich panels in the past decades [31,32,33,34,35,36]. To manufacture slim lightweight concrete (LWC) SCS panels resistant to impact and blast loads, connectors were proposed in previous studies. SCS panels with connectors showed promising behaviour when subjected to static and impact loads [37]. Connectors act in couples and interlock the concrete core in SCS panels. By this interlocking mechanism, the up-lift as well as local buckling of the steel faceplates are prevented. In SCS panel members with steel connectors, the transverse shear force is withstood by concrete core and connectors. Connectors act as shear links to avoid inclined shear cracks in the concrete core. Moreover, the tensile strength of the connectors contributes to the lateral shear resistance of the SCS panels. Another function of the connector is that their tensile strength can be used to reduce local buckling of the steel faceplates and avoid the rupture of steel plates from the concrete core. Local buckling of the steel plates can happen when the SCS panels are tested under compression, or the compression region of the steel plate is subjected to a bending load. A significant separation between the steel plate and concrete happens when a SCS panel is tested under a lateral impact or blast loading [37]. Therefore, connectors can sufficiently connect concrete core and steel plate, avoiding tensile separation and keeping the overall structural performance. As a result, the tensile resistance of connectors is a vital feature that will influence the structural behaviour of the SCS panels. This offers the motivation for the current investigation to assess the tensile strength of this novel type of connectors.

Punching shear rupture can happen in the steel plates of SCS members under different loads, and so many experimental studies have been done by Huang et al. [38] and Sohel and Liew [39] on this mechanism. However, these researchers only reported limited evidence on the SCS sandwich panels. In these panels, after flexural yielding, a membrane action is developed in the slab due to the effectiveness of connectors in maintaining composite action that further increases its load-carrying capacity after flexural yielding. The main benefit of the connectors is providing the shear capacity to restrict the shear cracks resulting from the shear force between the concrete core and the steel plates under an external lateral load. Lately, diverse sorts of connectors have been invented to enhance the shear performance of SCS members [40,41]. Furthermore, different types of the concrete core, such as LWC and high-strength concrete, were assessed in previous studies [42]. In the initial investigations, normal weight concrete (NWC) was used to produce the concrete core [43]. After that, lightweight concrete (LWC) was used to decrease the weight of SCS members [44]. Sohel et al. [45] tested eight SCS slabs with J-shaped connectors and LWC core. Furthermore, the membrane effect after yielding was evaluated. The results demonstrated that the failure modes and cracks propagation in SCS slabs with J-hook connectors are very comparable to those of reinforced concrete (RC) slabs. In another research, Yan et al. [46] investigated the punching shear strength of LWC slabs and panels. They established a new model to estimate the shear capability of RC shells, but not valid for SCS shell members. Subsequently, a new formula was established to anticipate the bending and shear capability of SCS shells [47,48].

Nevertheless, there are inadequate design procedures to estimate the punching capacity of the SCS members and some standards such as Eurocode [49] and ACI 318 [50] have established general schemes for RC slabs. To enhance the behaviour of SCS panels in offshore structures, J-hook connectors were tested by Liew et al. [51]. According to this study, SCS sandwich panels with J-hook connectors exhibited outstanding performance under the impulse and long-term loads. In 2009, Liew and Sohel [52] proposed a new technique to design SCS sandwich panels with LWC core. In their study, J-hook connectors were utilized to raise the shear capability. The obtained experimental outcomes showed that the J-hook connector is an efficient element to withstand the shear stress between steel plates and concrete core. In 2015, Yan et al. [53] measured the bending performance of SCS LWC beams experimentally and numerically. The results of the failure mode and shear strength of structures show the influence of the thickness of the steel skin shell, curvature, spacing of the connectors, depth of the cross-section, the strength of the concrete core, and boundary conditions on the ultimate strength behaviour of the curved SCS sandwich beam. In 2011, Leekitwattana [54] employed corrugated-strip connectors (CSC) to produce SCS panels. These connectors are placed normally to the inclined crack of concrete. Consequently, a limitation should be defined for the concrete core’s thickness. The stud-bolt connector is one of the cheapest and simplest types of connectors. In 2017, Yousefi and Ghalehnovi [55] studied the impact of one-end welded corrugated-strip connectors on the shear behaviour of SCS panels. The specimens were produced with different connectors’ angles and tested under the push-out test. The results indicated that, by increasing the connectors’ strength, the shear strength considerably increased. In another study, Yousefi and Ghalehnovi [56] proposed a FEM scheme to foresee the shear capability of SCS panels with one-end welded corrugated-strip connectors. In that study, a formula was proposed to control the interlayer performance of a double-skin SCS structure.

## 2. Research Significance

Based on the results of the previous studies, using diverse connectors enhances the shear capability of SCS panels, but in this study, a new type of stud-bolt connector was applied, which is cheap and easy to use. To manufacture the previous connectors, some equipment such as a welding motor and electrode is needed to weld the connectors to the steel faceplates. Additionally, the heat produced to weld is going to deform the molecular structure of the steel plates and results in a reduction of the strength of plates and an increase of stress concentration in the welding points. In addition, to present a new low-cost connector, CRA were sorted from a demolished old building and used as a substitute of CNA at 100% in terms of weight to consider solving environmental problems. According to previous investigations, the use of CRA decreased the load-bearing capacity of concrete members. Therefore, to mitigate the negative effect of CRA on the load-carrying of SCS panels, SF were added to specimens. Furthermore, a new model is presented to estimate the shear capability of SCS panels with stud-bolts connectors. Figure 1 shows the experimental program and novelty carried out in this study.

## 3. Materials and Specimens’ Specifications

### 3.1. Steel Plates

Square steel faceplates with 300 mm sides and 6 mm thickness were cut at the factory and holes were created using computer numerical control (CNC), as seen in Figure 2. In addition, in order to determine the characteristics of the steel faceplates, three specimens were tested under the directional tensile test according to ASTM A517 [57], and the average properties were considered and those of the steel plates and presented in Table 1.

### 3.2. Bolts and Nuts

In this study, bolts of three diameters (8 mm, 10 mm and 12 mm) were used. To manufacture, the SCS sandwich panels, two bolts and four nuts were used. The mechanical characteristics of the bolts are presented in Table 2. The shear bearing capacity of the bolts with diameters of 8 mm, 10 mm and 12 mm was obtained (30,000 N, 55,000 N and 58,000 N, respectively), according to the properties presented in Table 2. There were two bolts’ spacings: 100 mm and 150 mm. Figure 3 presents the reinforced plates with stud-bolt connectors. The bolts were selected according to ASTM C293 [58,59,60].

### 3.3. Concrete Core

The spacing between plates is filled with concrete to make the concrete core, where both NWC and steel fibres concrete (SFC) were used. Furthermore, two thicknesses of the concrete core were considered: 80 mm and 100 mm. The width and length of the specimens were 300 mm and 250 mm, respectively, according to the standards [58,59,60,61,62,63,64,65,66,67,68,69,70,71,72,73,74]. The concrete mixes’ characteristics are represented in Table 3. In addition, SF are used at 1% (by volume). In addition, CRA were used as a substitute of CNA at 100% in terms of weight. To evaluate the concrete’s compressive strength, three cylindrical specimens with 150 mm diameter and 300 mm height were produced and tested under a hydraulic jack. For each mix, the average compressive strength of three samples was considered [62,63,64,65,66]. The results are explained in Table 4. The specimens were tested after 28 days. Figure 4 presents a filled specimen.

### 3.4. Steel Fibres

In this study, two bent ends are used as illustrated in Figure 5. The tensile strength, elasticity modulus and failure strain of the fibres are 200 GPa, 2 GPa and 3%, correspondingly. Additionally, the length and equivalent diameter of SF are 60 mm and 0.9 ± 0.03, respectively. SF are employed in order to produce concrete at 1% volumetric content.

### 3.5. Coarse Recycled Aggregate

In this study, CRA were sourced from building demolition and replaced CNA at 100% weight ratio. The physical and chemical characteristics of CRA are provided in Table 5 and Table 6, respectively. In addition, in Figure 6, the chemical composition of CRA achieved by XRD is illustrated. There, each component is drawn with a colour spectrum.

### 3.6. Specimens’ Properties

In this study, 24 SCS sandwich panels were manufactured and tested. In these specimens, the diameter of bolts, bolts’ spacing, thickness and type of concrete core were variable. The geometric characteristics of the specimens are illustrated in Table 7.

In the designation of the specimens in Table 7, N, F, D, T and S specify normal concrete, SFC, the bolts’ diameter, the thickness of the concrete core and spacing of the bolts.

## 4. Test Setup

Specimens were produced and measured under a hydraulic jack for external pressure loading, as represented in Figure 7a. To evaluate deformations, two LVDTs were set up at the top and bottom of the specimens. The test was done under displacement control circumstances at a 0.5 mm rate, and the stoppage condition was set to be a failure. The purpose of this test was to determine the ultimate load and slip of the sandwich panels. Additionally, Figure 7b shows the force distribution in the specimens under lateral load.

## 5. Results and Discussion

To consider the impact of the stud-bolt connectors on the shear behaviour of SCS, with two sandwich panels, different outcomes were obtained and discussed in the following sections.

### 5.1. Shear Behaviour and Failure Modes

The failure mode is an important factor to analyse the performance of SCS panels under different loads. In Table 8, the failure mode of the specimens is identified. Furthermore, in Figure 8, the ultimate loading capability of specimens is demonstrated. It should be stated that the loading was performed over the thickness of specimens in the lateral and shear direction, as illustrated in Figure 7b. As seen in Table 8, SF significantly increased the shear strength of the specimens, especially when 12 mm diameter bolts were used. The main reason for improving the shear strength of SCS sandwich panels due to using SF could be attributed to the bridging role of fibres. Adding SF keeps particles together and increases the stiffness of concrete paste. In addition, under the shear stress, stress is transferred through cracks using SF, which leads to increasing the concrete core’s resistance exposed to a shear load. Additionally, the bridging role of SF led to a significant reduction in cracks width, which is one of the main reasons for bolt failure and results in reaching the highest load-bearing capacity of SCS sandwich panels.

Moreover, the bearing capacity increased by increasing the concrete core’s thickness. Furthermore, simultaneously increasing both the bolts’ spacing and core thickness had a negative effect on the maximum strength of SF reinforced concrete specimens when bolts with a higher diameter (>8 mm) are used, while, in specimens with 8 mm bolts’ diameter, increasing both the bolts’ spacing and core thickness causes the ultimate shear strength to rise considerably. Furthermore, in specimens with no SF, the failure modes changed from localized failure at the bottom end of the bolts to concrete core fracture by increasing the diameter of the bolt above 8 mm. The same results were reported by previous studies for other types of connectors which confirms the results presented in this study [68,69,70]. In contrast, the mode of failure was crushing of the concrete core as a result of bolts failure when SF was used because adding SF increases the compressive, tensile and shear strength of concrete and concentrates more stress on the bolts. These changes are visible in Figure 8. Increasing the bolts’ diameter resulted in concrete core crushing and fracture mode in the specimens. Conversely, the concrete strength increased when SF was used. Therefore, in specimens with lower bolt diameter, the failure mode occurred by bolts failure; however, failure of the concrete core occurred by increasing the bolts’ diameter. According to Figure 8, in specimens with 8 mm bolts’ diameter, increasing both the core thickness and bolts’ spacing increased the shear strength of the panels, while in specimens with 10 mm and 12 mm bolts’ diameter, the maximum shear strength was achieved when the core thickness and bolts’ spacing were 100 mm and 100 mm, respectively. The reason for this phenomenon is less contribution of the bolts with a diameter of 8 mm to shear strength and a significant contribution to the shear force of the concrete’s strength; however, by increasing the bolts’ diameter (10 mm and 12 mm), the role of bolts in providing shear strength increases, which resulted in better shear performance for specimens for lower bolts’ spacing. Additionally, the improvement of the maximum shear strength of specimens with 100 mm bolts’ spacing from raising the core thickness is more significant in those with 150 bolts’ spacing.

Figure 9 illustrates the mode of failure samples. In NWC with 8 mm bolts’ diameter, the cohesion between concrete core and faceplates dropped and the plates detached from the concrete core by increasing the load. Besides, the concrete core failed when the bolts’ spacing increased because the confinement of concrete between bolts drops as a result of increasing the bolts’ spacing. The concrete core was completely fractured by raising the bolts’ diameter. In addition, the crack width increased when the bolts’ diameter went up. Furthermore, cracks width decreased by using SF due to the bridging role of fibres that kept particles together in the concrete paste and increased the strength of the concrete matrix, and the ultimate bearing capacity significantly improved. Therefore, detachment between the concrete core and plates occurred by increasing the load. In addition, the crack width decreased, and cracks propagated more.

In order to consider the impact of bolts’ diameter, the load-slip relationship of specimens with the same concrete core’s thickness and bolts’ spacing was evaluated, as illustrated in Figure 10 and Figure 11 for NWC and SFC, correspondingly. The ultimate shear strength substantially increased when bolts of 12 mm diameter were used, but there is no significant variance between the ultimate shear strength of specimens made with bolts of 8 mm and 10 mm diameter; however, the shear capacity of specimens with 10 mm bolts’ diameter is slightly higher. This could be attributed to increasing the area of bolts, which leads to reducing the stress value over bolts’ diameter and is the main reason for improving the shear strength of the specimens. There are two peaks in the load-slip behaviour of some specimens (Figure 10a and Figure 11a). According to these figures, the shear strength of the specimens increased and then slightly dropped as a result of the concrete core’s fracture and then went up and declined again after the second peak resulting from bolts’ failure and detachment from the steel plate face. Additionally, there is no relevant difference in slip at ultimate strength by increasing the concrete core’s thickness.

As shown in Figure 11, in specimens with 80 mm core thickness and 100 bolts’ spacing, increasing the bolts’ diameter to 10 mm and 12 mm substantially increased the maximum slip by about 475% and 625% for NWC, and 233% and 283% for SFC, respectively. This could be attributed to increasing the deformation of the bolts with an increase of the bolts’ diameter. According to Figure 10 and Figure 11, the initial stage of the loading curves was linear-elastic followed by a sudden drop of the force due to tension cracking in normal concrete core specimens. In contrast to SFRC cored specimens, only a single crack was observed in the mid-section of the specimen indicating that the sandwich plate failed by core tension, and possibly core/face debonding in certain areas, which led to a sudden drop of the applied force after the maximum load-bearing capacity. Very similar results were reported by previous investigations for CNA concrete specimens [68]. On the other hand, in specimens with 150 mm bolts’ spacing, increasing the bolts’ diameter considerably increased the slip at ultimate strength; however, in NWC, the slip at ultimate strength dropped by increasing the bolts’ diameter to more than 10 mm while the slip at ultimate strength of SFC specimens improved by increasing the bolts’ diameter to more than 10 mm.

Moreover, specimens with 100 mm bolts’ spacing and 8 mm, 10 mm and 12 mm bolts’ diameter, the maximum shear strength is considerably increased by adding SF by approximately 40%, 11% and 44%, respectively, in specimens with 80 mm core thickness while in specimens with 100 mm core thickness, the improvement was about 33%, 25% and 30%, respectively. Furthermore, in specimens with 150 mm bolts’ spacing and 8 mm, 10 mm and 12 mm bolts’ diameter, the maximum shear capability improved by adding SF by about 17%, 13% and 20%, respectively, in specimens with 80 mm core thickness, while in specimens with 100 mm core thickness this improvement was nearby 25%, 26% and 20%, respectively. Therefore, the impact of SF was boosted for higher diameter of the bolts. Consequently, using SF considerably improves the ultimate shear strength.

In order to investigate the impact of concrete core’s thickness on the performance of SCS sandwich panels, the influence of the type of concrete core (NWC and SFC) on the load-slip behaviour of specimens was assessed, as seen in Figure 12 and Figure 13, respectively.

In NWC, there is a significant difference between the slip values at ultimate shear strength. Additionally, increasing the concrete core’s thickness did not improve the maximum shear strength of the specimens when the bolts’ spacing is 100 mm and 8 mm and 10 mm bolts’ diameter are used; however, it improves the maximum strength of 12 mm bolts’ diameter reinforced SCS by about 36% (Figure 12a). Thus, increasing the concrete core’s thickness is not a good way to increase the shear strength of the SCS when bolts with a smaller diameter (≤10 mm) are used. The same trend can be observed when the bolts’ spacing is 150 mm (Figure 10b). Generally, raising the core thickness is not an effective way to improve the ultimate shear strength of the SCS panels; however, raising the bolts’ diameter could be a useful manner to improve the shear performance of composite panels. Oppositely, in SFC samples, the maximum shear strength went up with the increase of concrete core’s thickness when the bolts’ spacing was 100 mm, while by increasing the bolts’ spacing to 150 mm the ultimate shear strength is not enhanced by increasing the core thickness and the slip at ultimate strength declined as well (Figure 13a,b). As a result, it is recommended that, in order to improve the shear capacity of SCS sandwich panels, increasing the bolts’ diameter and using SF play an important role; however, raising the bolts’ spacing and the concrete core’s thickness did not have a considerable influence on the maximum shear strength. In Figure 14 and Figure 15, the impact of SF on both the ultimate strength and slip at ultimate strength was studied. As illustrated, SF significantly improve the performance of SCS sandwich panels.

### 5.2. Ductility

In this section, the ductility index of specimens was evaluated. This index shows the deformation capability of specimens and is defined as the ratio between ultimate deformation and the corresponding displacement at yielding. Therefore, the value of ductility could be calculated using the following formula, as illustrated in Figure 16. The achieved experimental consequences of ductility are presented in Table 9 and Figure 17. As shown in Figure 17, raising the diameter of bolts and using SF resulted in considerable improvement of the value of ductility and deformation of specimens. The main reason for improving the ductility of SCS sandwich panels with SF could be attributed to increasing the strength of CRA concrete matrix obtained from the transferred stress through cracks using SF.

The yield point is defined as the point where the behaviour of specimen starts to be nonlinear. The ductility index is a qualitative parameter that depends on the deformation of specimens and brittle or ductile failure mode of the concrete members. Therefore, the exact value of ∆y is not of major importance. As seen in Table 9 and Figure 17, in addition to SF, increasing bolts’ diameter plays an appropriate role to improve the ductility and deformation of specimens and prevents brittle shear failure of specimens.
(1)$$i=\frac{\Delta_{u}}{\Delta_{y}}$$

### 5.3. Compression with Other Types of Connectors

In order to show the performance of the proposed connectors in this study, the results were compared with those presented in previous studies. Liew et al. [67] proposed a J-hook connector to raise the shear strength of SCS panels. In another research, Yousefi and Ghalehnovi [55] proposed one-end welded corrugated-strip connectors in SCS members. In Figure 18, the structure of J-hook and welded corrugated-strip connectors are illustrated. Moreover, the characteristics of the specimens are represented in Table 10. Therefore, the compression results are illustrated in Figure 19 and Figure 20, and listed in Table 11. Even though not all the characteristics of the specimens from previous researches are the same as those from this study, they are similar, which allows an acceptable comparison. Stud-bolts connectors have been used to improve the shear of composite beams and shear walls, but there is no study about the shear behaviour of SCS sandwich panels with stud-bolts, which is the novelty of this study.

According to Figure 19, the bearing capacity of the specimens significantly increased when stud-bolt connectors were used relative to those produced with J-hook connectors. This could be attributed to connecting two ends of the stud-bolt while in the J-hook connector only one end was welded to the plate. The concrete compressive strength of specimens produced using J-hook connectors and tested by Liew et al. [67] was almost twice of the one of those manufactured with stud-bolts connectors and tested in this study, but other parameters of specimens are similar.

Consequently, the compressive strength of the concrete core noticeably dropped when stud-bolt connectors were employed, which is the main advantage of stud-bolts connectors compared with the J-hook type. Besides, using SF enhanced the shear behaviour of the SCS members due to the bridging role of SF, which improved the shear resistance of the concrete matrix. The maximum shear capability of the specimens was also substantially improved when stud-bolts connectors were used compared to the specimens with welded end connectors (Figure 20). It should be stated that the yield strength of steel faceplates used by Yousefi and Ghalehnovi [55] to manufacture welded end connectors SCS sandwich structures was greater than that of the plates employed in this study, which is the foremost benefit of stud-bolts connectors. Therefore, the steel plated with lower yield strength could be used when stud-bolts connectors are used.

### 5.4. Suggested Load-Slip Behaviour Model for SCS Sandwich Panels with Stud-Bolt Connectors

In this subdivision, a new technical scheme is presented by using a regression technique using the experimental results. This method is mandatory to assess the load-slip relationship and estimating the maximum capability of SCS members with stud-bolt connectors. Previously, many formulas were proposed by Ollgaard et al. [31], Cederwall [68], Lorence and Kuica [69], Gattesco and Giuriani [32] and Xue et al. [70] to estimate the shear load-slip of SCS panels based on the experimental results as briefly presented in Table 12. The previously proposed models are appropriate tools to predict the shear behaviour of SCS sandwich panels for other types of connectors. Therefore, it is necessary to develop a model in order to estimate the shear strength of the SCS sandwich panels when stud-bolts are used.

In order to present a new technique to anticipate the shear behaviour of SCS panels, the bearing capability of specimens was normalized relative to P/Pu is the ultimate bearing capacity and can be determined based on the load-slip relationship. The result is presented in Figure 21. The normalized bearing capability (*P*/*Pu*) and slip (δ) are alienated into four categories (a, b, c and d) based on the type of concrete core and the diameter of bolts (Figure 21). According to Figure 22, the following equations estimate the shear performance of NWC SCS panels. Equations (2) and (3) differ in terms of the bolts’ diameter. The R^2^ values of Equations (2) and (3), 0.98 and 0.97 respectively, show a good fit with the experimental results.
(2)$$\frac{P}{P_{u}}=0.24ln\left(\delta \right)+0.64\;\;\;\;\;D\leq 8\;mm\;\;\;\;R2=0.98$$
(3)$$\frac{P}{P_{u}}=0.26ln\left(\delta \right)+0.42\;\;D\geq 10\;mm\;\;\;R2=0.97$$

For SFC specimens, a diverse scheme was proposed, as represented below. These formulas, with R^2^ values of 0.95 and 0.91, respectively, also show a good fit with the experimental consequences. They also differ in terms of the bolts’ diameter for SFC SCS panels. The comparison of the proposed formulas with the experimental results is illustrated in Figure 23.

When the diameter of bolts is up to 8 mm, Equations (2)–(5) can be used to forecast the shear capability of NWC and SFC SCS panels, correspondingly. Alternatively, Equations (4) and (5) can be employed when the diameter of bolts is equal or greater than 10 mm. Besides, in Figure 24, the presented formulas were compared with the models proposed for previous techniques. This figure demonstrates the higher accuracy of the presented model in this study relative to the previous ones. Previous models were based on one equation that is not useful for the wide-range of SCS panels. In this research, a two-dimensional equation is proposed that covers a wide-range of specimens based on the bolts’ diameter, which shows the high accuracy of the model proposed in this study.
(4)$$\frac{P}{P_{u}}=\;0.5\delta^{0.53}\;\;\;\;D\leq 8\;mm\;\;\;R2\;=\;0.88$$
(5)$$\frac{P}{P_{u}}=\;0.19\delta^{0.8}\;\;D\geq 10\;mm\;\;\;\;R2\;=\;0.91$$

## 6. Conclusions

In this study, a new model of stud-bolt connector was presented. The effect of this connector on the shear capability of SCS panels was the object of this study. Additionally, the bolts’ diameter, concrete core’s thickness, bolts’ spacing and concrete type were the governing parameters. In total, 24 laboratory samples were produced and examined, and the shear performance of SCS panels was assessed. According to the obtained outcomes, the following conclusions could be discussed:
CRA could be used as an effective way to produce concrete members in order to decrease waste materials resources worldwide. In addition, the use of SF plays an effective role to mitigate the negative influence of high CRA incorporation on the structural performance of SCS sandwich panels;The new proposed stud-bolt connector is efficient to improve the shear strength of concrete sandwich panels, and this type of connectors are low-cost and easy to design compared with others;The failure mode was changed when the diameter of bolts is kept constant and SF were used. On the contrary, increasing the bolts’ spacing increases the deformation of the SCS panels when bolts with a higher diameter (>8 mm) are used while, in specimens with 8 mm bolts’ diameter, increasing both the bolts’ spacing and the core thickness causes the ultimate shear strength to rise considerably. Moreover, in specimens with 8 mm bolts’ diameter, increasing both the core thickness and the bolts’ spacing increases the shear strength of the panels while, in specimens with 10 mm and 12 mm bolts’ diameter, the maximum shear strength was achieved when the core thickness and bolts’ spacing are 100 mm and 10 mm, respectively. The reason for these phenomena is the low contribution of the bolts of 8 mm diameter to shear strength and significant increase of shear force with concrete strength; however, by increasing the bolts’ diameter (10 mm and 12 mm), the contribution of the bolts to shear strength increases, which resulted in better shear performance for specimens with less spacing;In specimens with no SF, the failure modes change from localized failure at the bottom end of the bolts to concrete core fracture by increasing the bolts’ diameter above 8 mm. Furthermore, the concrete core fractured when the bolts’ spacing increased because the confinement of concrete between bolts drops as a result of increasing bolts’ spacing. In contrast, the failure mode is crushing of the concrete core when SF are used;Increasing the bolts’ spacing causes failure of the concrete core. In addition, the concrete core completely fractures, and the crack width increases by increasing the diameter of the bolts. Alternatively, the crack width dropped and the maximum bearing capability increased by using SF because fibres play a bridge role to keep particles close to each other;The shear strength of the specimens increased and then slightly dropped as a result of the concrete core fracture and then raised and declined again after the second peak resulting from bolts’ failure and detachment from the steel plate face. Additionally, there is no relevant difference in slip at ultimate strength by increasing the concrete core’s thickness. Therefore, in specimens with 100 mm bolts’ spacing and 8 mm, 10 mm and 12 mm bolts’ diameter, the maximum shear strength considerably increased by adding SF by approximately 40%, 11% and 44% respectively in specimens with 80 mm core thickness, while in specimens with 100 mm core thickness the improvement was about by 33%, 25% and 30%, respectively. Furthermore, in specimens with 150 mm bolts’ spacing and 8 mm, 10 mm and 12 mm bolts’ diameter, the maximum shear capacity improved by adding SF by about 17%, 13% and 20% in specimens with 80 mm core thickness, respectively, while in specimens with 100 mm core thickness this improvement was nearby 25%, 26% and 20%, respectively;Increasing the concrete core’s thickness did not improve the maximum shear strength of specimen when the bolts’ spacing was 100 mm and 8 mm and 10 mm bolts’ diameter was used; however, it improved the maximum strength of 12 mm bolts’ diameter reinforced SCS by about 36%. Therefore, increasing the concrete core’s thickness is not a good way to increase the shear strength of SCS when bolts with a smaller diameter (≤10 mm) are used;In SFC samples, the maximum shear strength went up with the increase of concrete core’s thickness when the bolts’ spacing is 100 mm while, by increasing the bolts’ spacing to 150 mm, the ultimate shear strength was not enhanced by increasing the core thickness and the slip at ultimate strength declined. As a result, it is recommended, in order to improve the shear capacity of SCS sandwich panels, to increase the bolts’ diameter and use SF; however, increasing the bolts’ spacing and concrete core’s thickness did not have a considerable influence on the maximum shear strength;Increasing the bolts’ diameter plays an appropriate role to improve the ductility and deformation of specimens and prevents brittle shear failure of the specimens;The proposed model is highly accurate in the estimation of the shear behaviour of SCS panels with stud-bolt connectors and can be used for both NWC and SFC.

It is stressed that this research investigated the effect of bolts’ diameter, concrete core’s thickness, bolts’ spacing, RCA and SF on the performance of SCS sandwich panels using a novel proposed connector shape. In the experiments, SF and RCA contents were kept constant in all specimens. Therefore, it is recommended to measure the effect of various concrete, fibres and aggregate types as well as both coarse and fine aggregates on the performance of SCS sandwich panels with different kinds of connectors in future studies.

## Figures and Tables

**Figure 1 materials-14-05185-f001:**
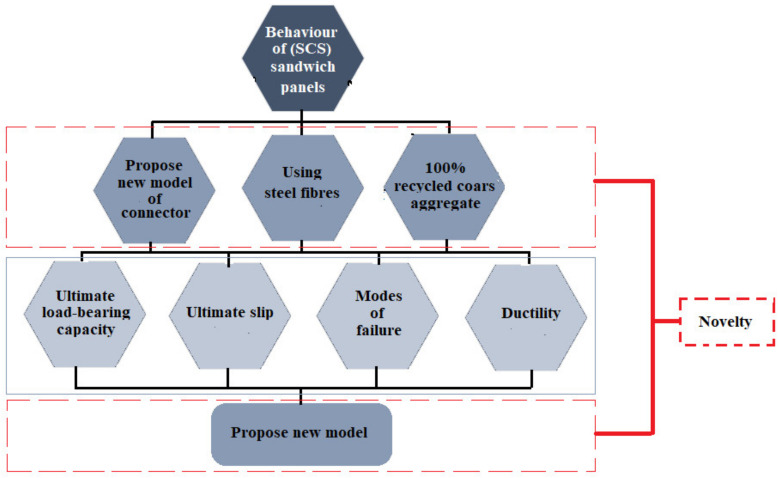
General flowchart of experimental program and novelty of this study.

**Figure 2 materials-14-05185-f002:**
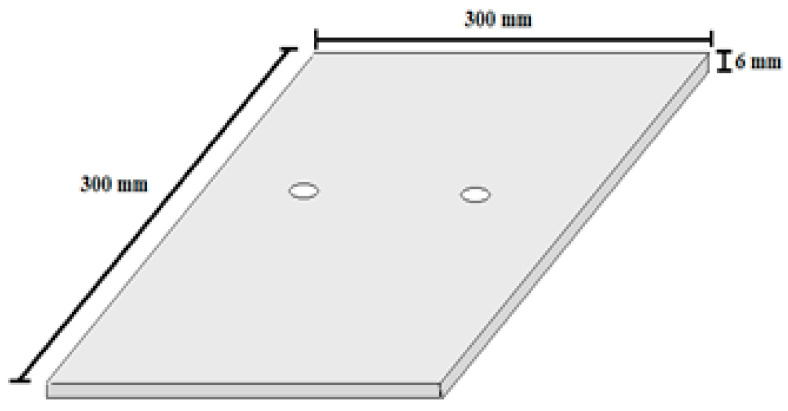
Geometry of the steel plates.

**Figure 3 materials-14-05185-f003:**
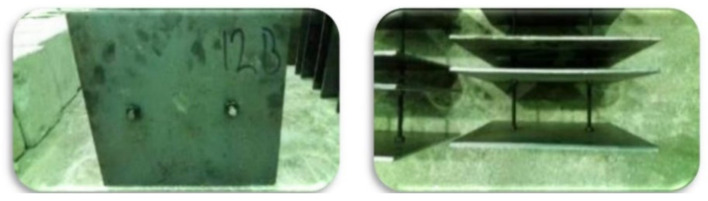
Reinforced plates with stud-bolt connectors.

**Figure 4 materials-14-05185-f004:**
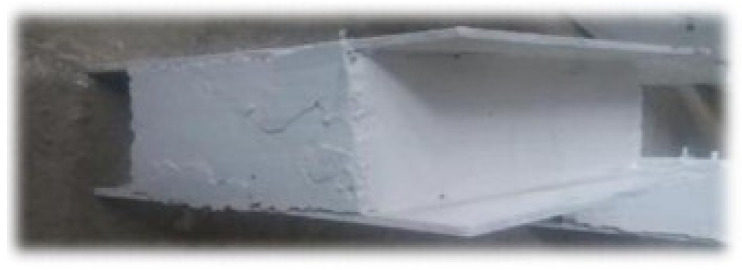
SCS sandwich panel.

**Figure 5 materials-14-05185-f005:**
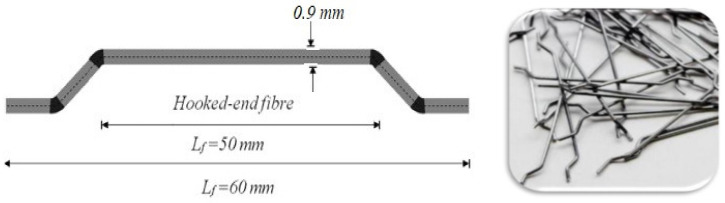
Steel fibres.

**Figure 6 materials-14-05185-f006:**
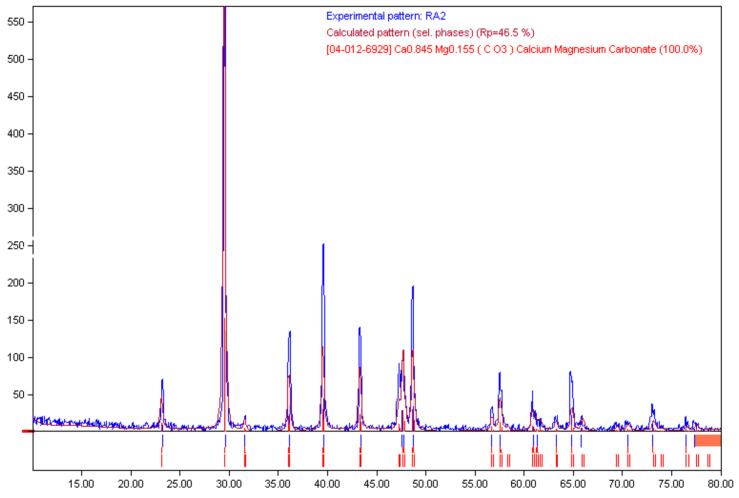
XRD patterns of CRA.

**Figure 7 materials-14-05185-f007:**
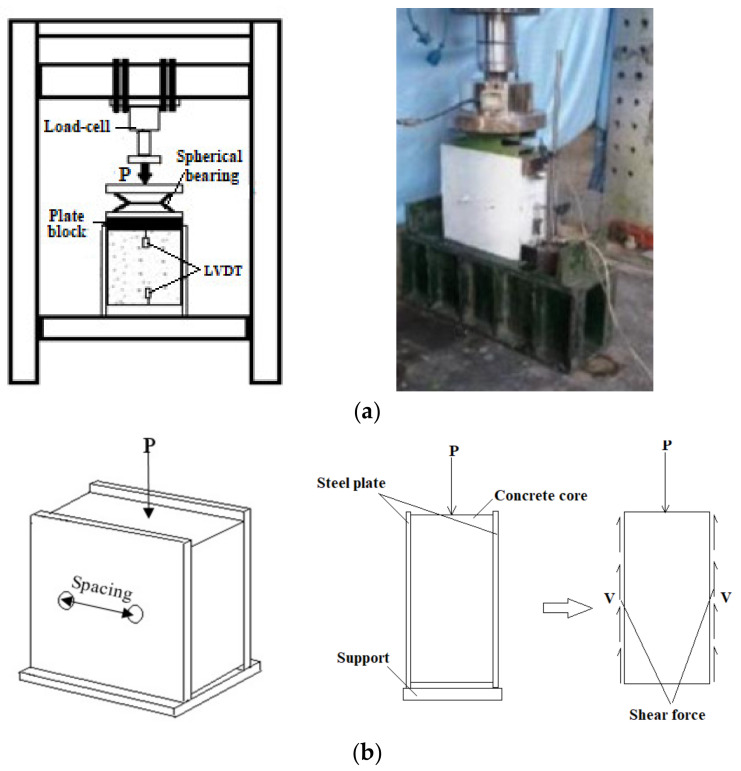
(**a**) Test setup and loading condition; (**b**) shear force distribution.

**Figure 8 materials-14-05185-f008:**
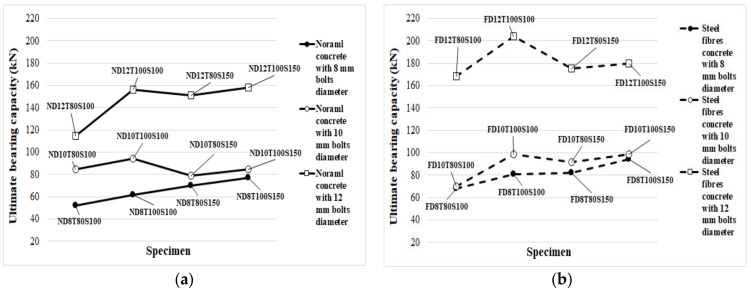
Impact of diverse factors on the critical bearing capacity: (**a**) NWC; (**b**) SFC.

**Figure 9 materials-14-05185-f009:**
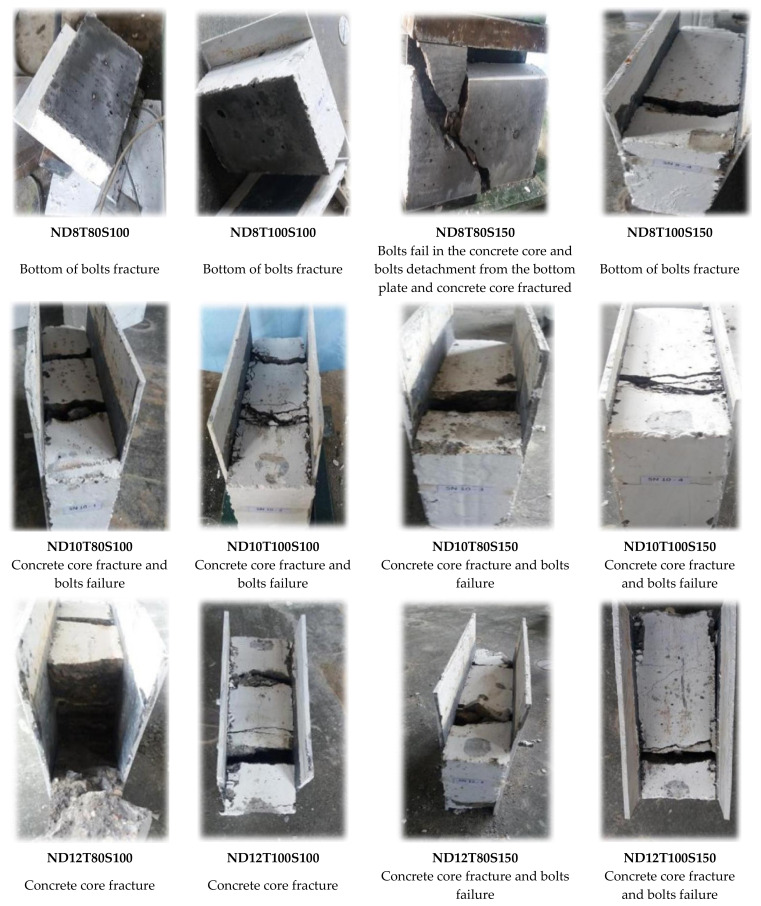
Typical static failure modes of specimens with and without SF.

**Figure 10 materials-14-05185-f010:**
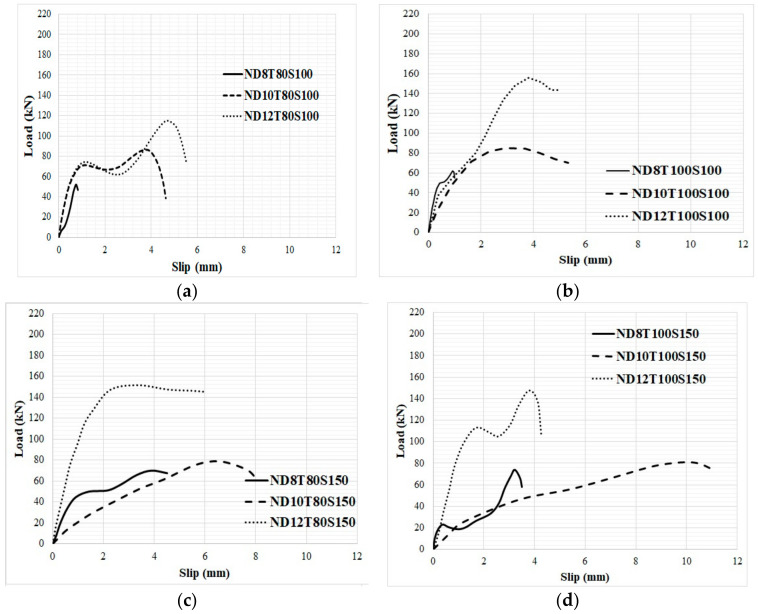
Impact of bolts’ diameter on the load-slip behaviour of specimens with NWC. (**a**) 8 mm bolt’s diameter with 100 mm spacing and 80 mm core thickness, (**b**) 8 mm bolt’s diameter with 100 mm spacing and 100 mm core thickness, (**c**) 8 mm bolt’s diameter with 150 mm spacing and 80 mm core thickness and (**d**) 8 mm bolt’s diameter with 150 mm spacing and 100 mm core thickness.

**Figure 11 materials-14-05185-f011:**
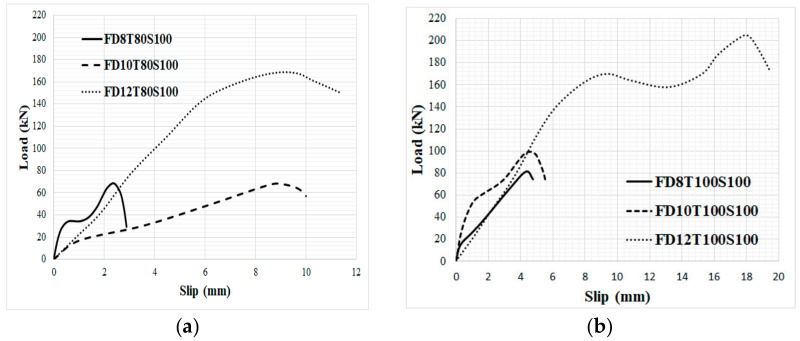
Impact of bolts’ diameter on the load-slip behaviour of specimens with SFC. (**a**) 100 mm spacing and 80 mm core thickness, (**b**) 100 mm spacing and 100 mm core thickness, (**c**) 150 mm spacing and 80 mm core thickness and (**d**) 150 mm spacing and 100 mm core thickness.

**Figure 12 materials-14-05185-f012:**
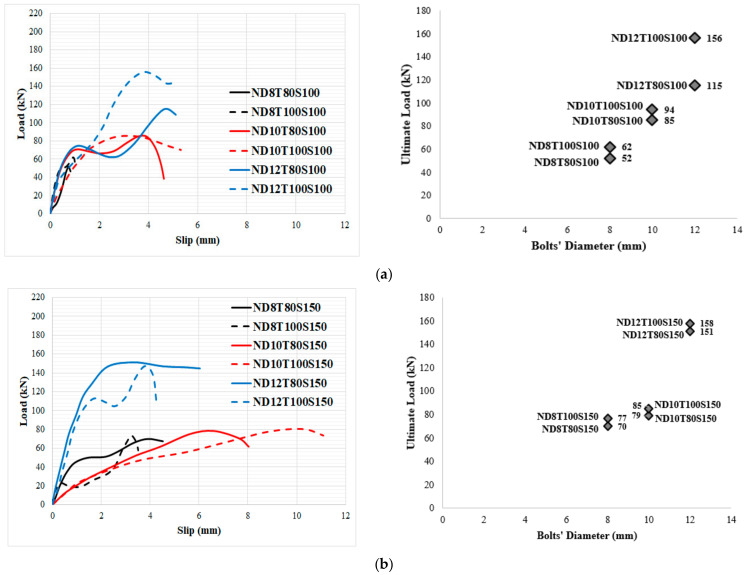
Impact of concrete core’s thickness on the performance of SCS panels with no SF. (**a**) 100 mm bolts spacing and (**b**) 150 mm bolts spacing.

**Figure 13 materials-14-05185-f013:**
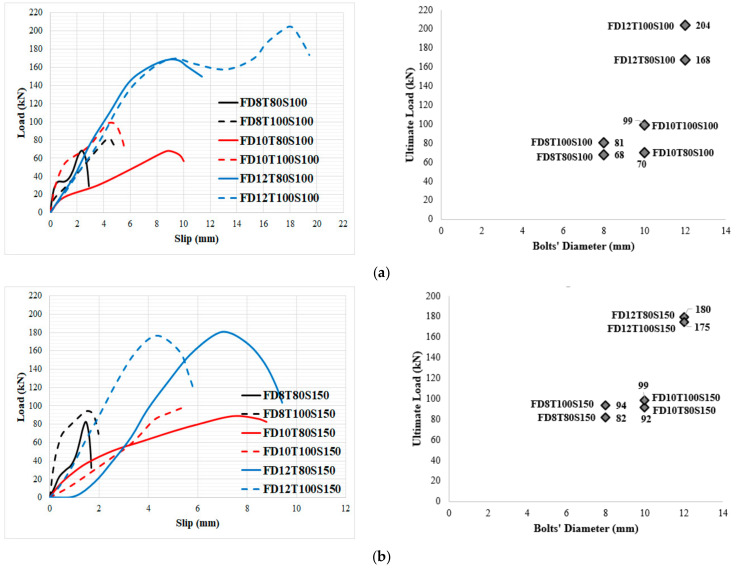
Impact of concrete core’s thickness on the performance of SCS panels with SF. (**a**) 100 mm bolts spacing and (**b**) 150 mm bolts spacing.

**Figure 14 materials-14-05185-f014:**
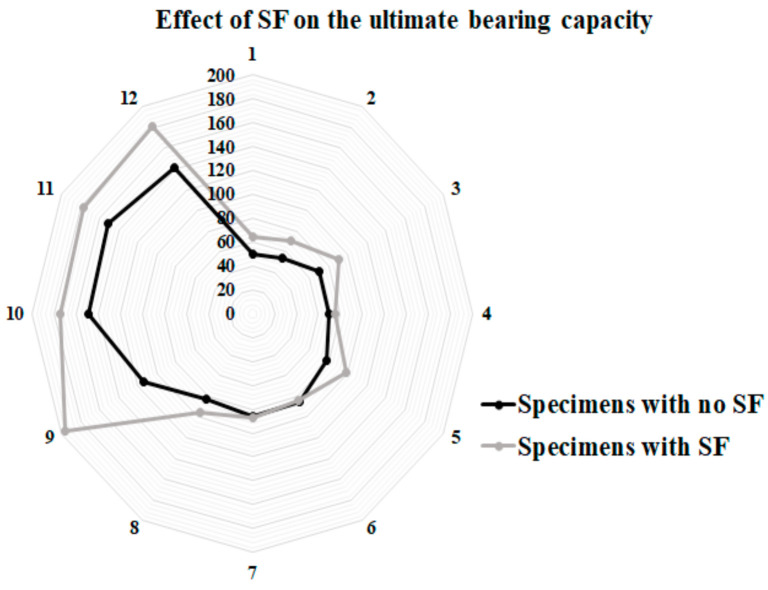
Impact of SF on the ultimate bearing capacity of specimens.

**Figure 15 materials-14-05185-f015:**
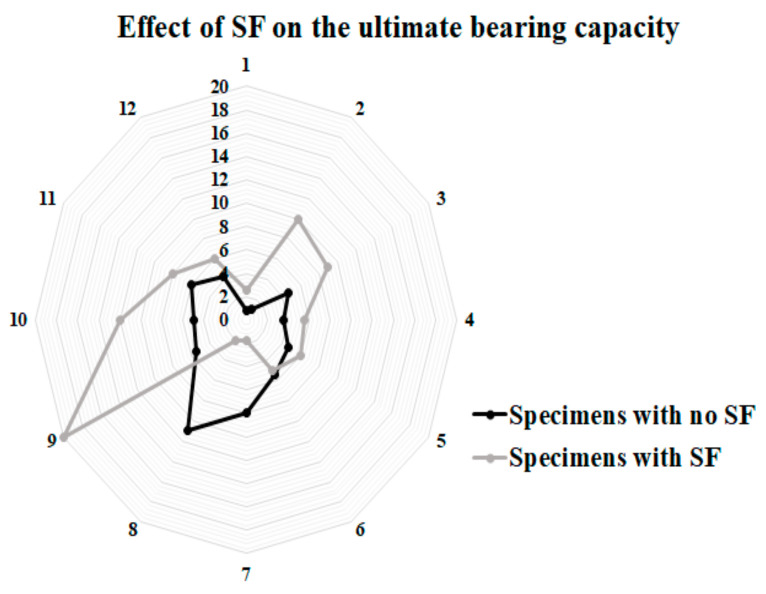
Impact of SF on the maximum slip of specimens.

**Figure 16 materials-14-05185-f016:**
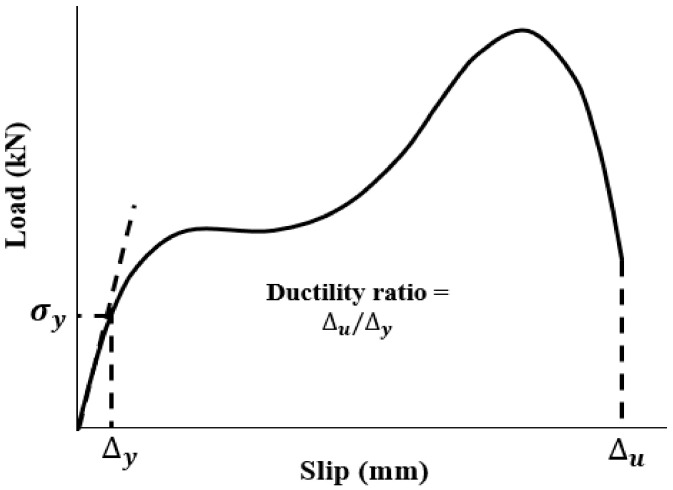
Ductility ratio.

**Figure 17 materials-14-05185-f017:**
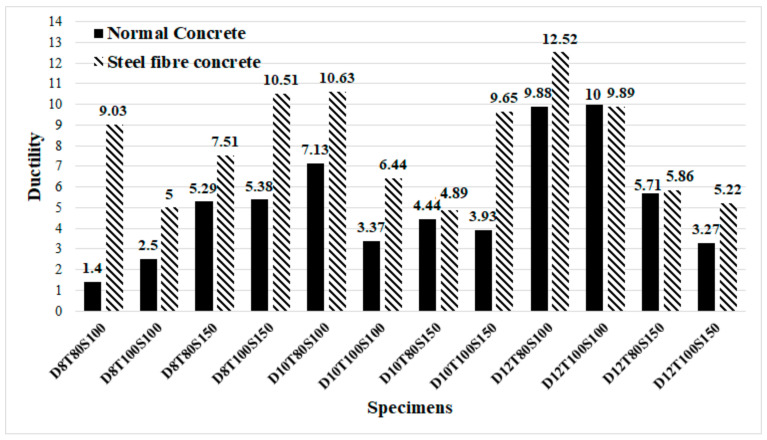
Ductility index of SCS sandwich panels.

**Figure 18 materials-14-05185-f018:**
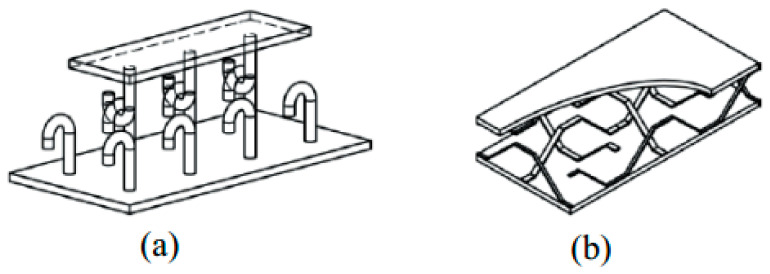
SCS sandwich constructions based on shear connector shape. (**a**) J-hook connectors and (**b**) welded end connectors.

**Figure 19 materials-14-05185-f019:**
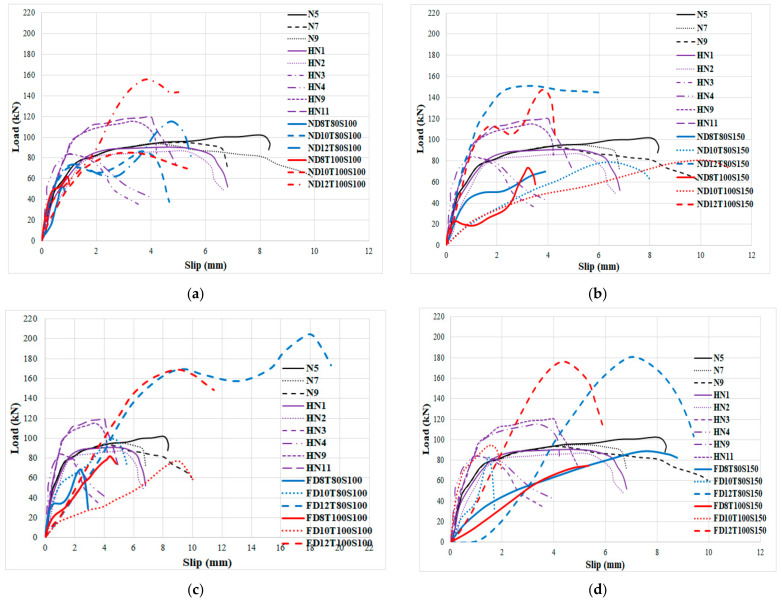
Compression between the proposed connector in this study and J-hook connectors.(**a**) normal concrete with 100 mm bolts’ spacing, (**b**) normal concrete with 150 mm bolts’ spacing, (**c**) fibre-reinforced concrete with 100 mm bolts’ spacing and (**d**) fibre-reinforced concrete with 150 mm bolts’ spacing.

**Figure 20 materials-14-05185-f020:**
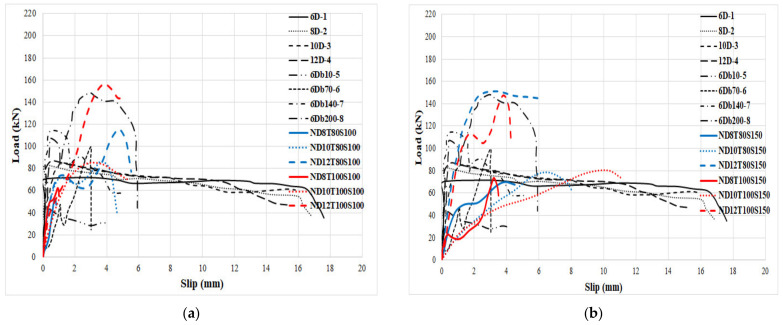
Compression between the proposed connector in this study and welded end connectors. (**a**) normal concrete with 100 mm bolts’ spacing, (**b**) normal concrete with 150 mm bolts’ spacing, (**c**) fibre-reinforced concrete with 100 mm bolts’ spacing and (**d**) fibre-reinforced concrete with 150 mm bolts’ spacing.

**Figure 21 materials-14-05185-f021:**
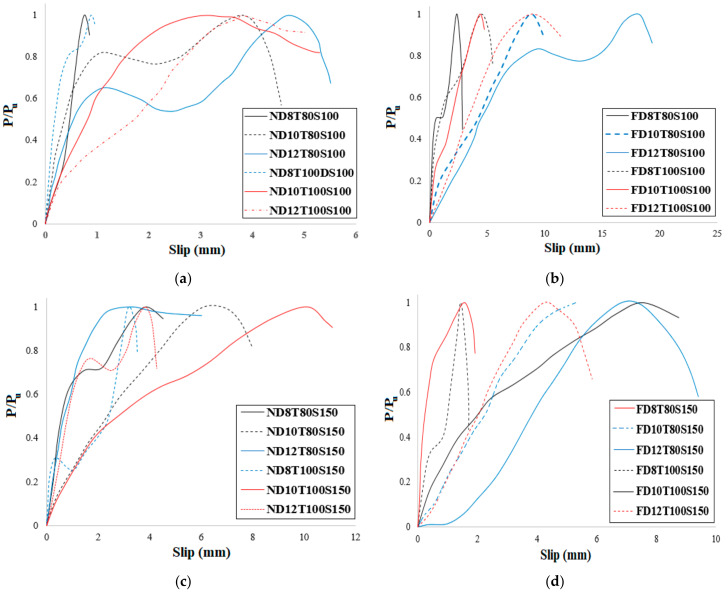
Normalized load-slip relationship of SCS specimens. (**a**) normal concrete with 100 mm bolts’ spacing, (**b**) fibre-reinforced concrete with 100 mm bolts’ spacing, (**c**) normal concrete with 150 mm bolts’ spacing and (**d**) fibre-reinforced concrete with 150 mm bolts’ spacing.

**Figure 22 materials-14-05185-f022:**
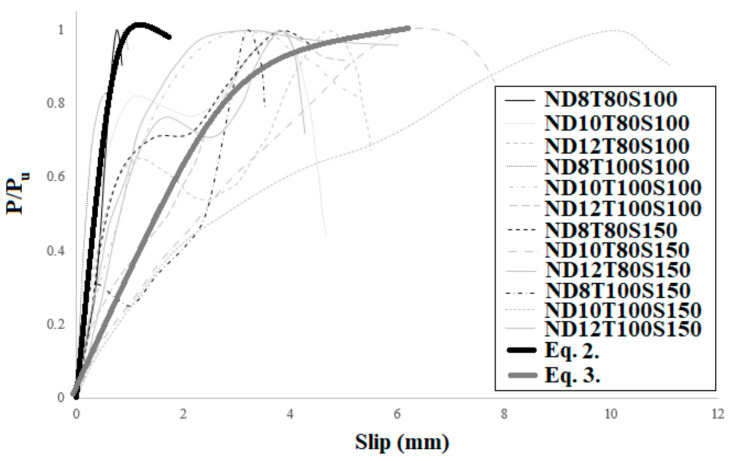
Comparison between the normalized load-slip relationship of the test results and the proposed model for NWC SCS panels with stud-bolt connectors.

**Figure 23 materials-14-05185-f023:**
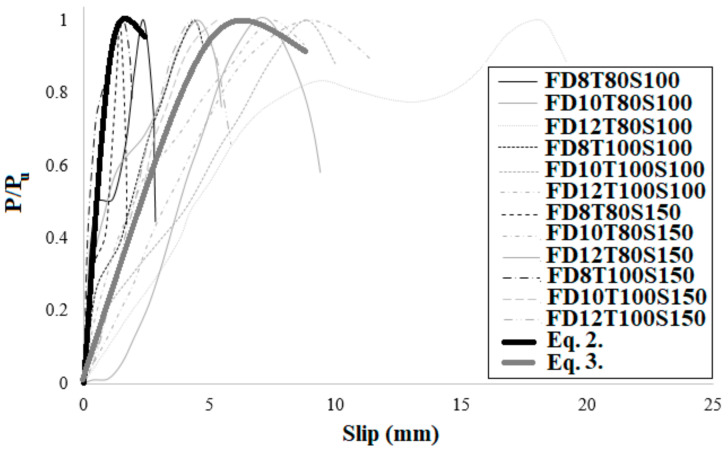
Comparison between the normalized load-slip relationship of the test consequences and the proposed technique for SFC SCS panels with stud-bolt connectors.

**Figure 24 materials-14-05185-f024:**
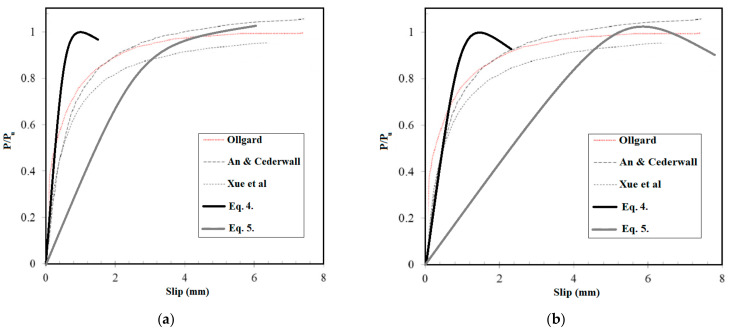
Comparison between the presented scheme and the proposed previous equations: (**a**) NWC (**b**) SFC.

**Table 1 materials-14-05185-t001:** Properties of steel plates with 6 mm thickness.

Materials	Yield Stress (MPa)	Ultimate Stress (MPa)	Modulus of Elasticity (GPa)	Ultimate Strain
Steel plate	283	491	201	0.0024
Coefficient of variation (%)	0.7	0.8	0.4	4.0

**Table 2 materials-14-05185-t002:** Mechanical characteristics of the bolts.

Service Classes										
ASTM C293 [58]		B5	B6	B7	B8	B8C	B8M	B8T	B16	B7M
Chemical Analysis										
Carbon		>0.15	<0.20	0.4–0.15	>0.10	>0.10	<0.10	<0.10	0.35–0.45	0.40–0.50
Manganese		<15	<2	0.70–1	<3	<3	<3	<3	0.50–0.75	0.70–1
Phosphorous mud		0.05	0.05	0.05	0.05	0.05	0.05	0.05	0.05	0.05
Sulphur mud		0.04	0.09	0.05	0.09	0.09	0.09	0.09	0.05	0.05
Silicon		<1	<1	0.20–0.35	<1	<1	<1	<1	0.20–0.35	0.25–0.35
Nickle		–	–	–	8–20	9–13	10–14	9–22	–	–
Chromium		4.5–6.5	12–14	0.85–1.2	5–20	15–20	15–20	15–20	0.8–1.2	0.85–1.15
Molybdenum		0.05–0.7	-	0.2–0.3	–	–	2.5–3.5	–	0.55–0.70	0.2–0.3
Vanadium		–	–	–	–	–	–	–	0.3–0.4	–
Tanium mini		–	–	–	–	–	–	6	–	–
Columbium + Titanium		–	–	–	–	–	–	–	–	–
Tensile Requirements										
Minimum tensile strength	Lbs/psi $\mathrm{kg}/{\mathrm{mm}}^{2}$	100.0	110.0	125.0	75.0	75.0	75.0	75.0	125.0	100.0
		70.5	77.5	88.0	53.0	53.0	53.0	53.0	88.0	70.5
Minimum yield strength	Lbs/psi $\mathrm{kg}/{\mathrm{mm}}^{2}$	80.0	85.0	105.0	50.0	30.0	30.0	30.0	105.0	80.0
		56.5	60.0	74.0	60.0	21.0	21.0	21.0	74.0	50.5
Elongation in 2 inches (%)		16.0	15.0	16.0	55.0	30.0	30.0	30.0	18.0	18.0
Reduction of area (%)		50.0	50.0	50.0	50.0	50.0	50.0	50.0	50.0	50.0
Internal Molecules Equilibrium										
AlSi		501	410	4140–4142	304	347	321	316	–	4142–4145
AFNOR		Z12CO5	Zr12cr13	4co4	Z6CN	Z6CNN	–	Z6CND	40CDV 4.06	42CD4
WERISTOFF		12 crMo 19.05	X10cr13	4crMo4	XSCNi 18.09	X10CNiNb 18.90	–	25cr NiMO 18.10	40crMoN 48	42CrMo4
B. S		15.06–625	15.06–713	15.06–624Gr.A	15.06–801Gr.A	–	–	150.6–845	150.6–661	150.6–62 GrA
**Recommended Temperature Range (°C)**										
Minimum		–	–	−45	−198	−198	−198	−198	−129	–
Maximum		–	–	−48.2	67	675	675	67	575	–

**Table 3 materials-14-05185-t003:** Concrete mix composition (kg/m^3^).

Materials	Water	Artificial Sand	Natural Sand	Recycled Coarse Gregates	Cement	Weight
Concrete	200	450	630	720	400	2400

**Table 4 materials-14-05185-t004:** Compressive strength of the various mixes (MPa).

NWC		SFC	
Compressive strength	Average strength	Compressive strength	Average strength
29.6	30.6	34.9	33.9
31.4		34.3	
30.5		32.6	

**Table 5 materials-14-05185-t005:** Physical properties of CRA.

Apparent Density (g/cm^3^)	Bulk Density (g/cm^3^)	Water Absorption (wt%)	Crushing Index (%)	Porosity (%)
2.66	2.56	1.519	46.1	3.76

**Table 6 materials-14-05185-t006:** Chemical properties of CRA.

Chemical Composition	CRA
Ca Mg(CO_3_) (%)	100
Overall diffraction profile (%)	100
Background radiation (%)	25.12
Diffraction peaks (%)	74.88
Peak area belonging to selected phases (%)	47.15
Peak area of Phase A (calcium magnesium carbonate)	47.15

**Table 7 materials-14-05185-t007:** Compressive strength of the various mixes (MPa).

Specimen	Bolts’ Diameter (mm)	Thickness of the Concrete Core (mm)	Bolts’ Spacing (mm)	Specimen	Bolts’ Diameter (mm)	Thickness of the Concrete Core (mm)	Bolts’ Spacing (mm)
ND8T80S100	8	80	100	FD8T80S100	8	80	100
ND8T100S100	8	100	100	FD8T100S100	8	100	100
ND8T80S150	8	80	150	FD8T80S150	8	80	150
ND8T100S150	8	100	150	FD8T100S150	8	100	150
ND10T80S100	10	80	100	FD10T80S100	10	80	100
ND10T100S100	10	100	100	FD10T100S100	10	100	100
ND10T80S150	10	80	150	FD10T80S150	10	80	150
ND10T100S150	10	100	150	FD10T100S150	10	100	150
ND12T80S100	12	80	100	FD12T80S100	12	80	100
ND12T100S100	12	100	100	FD12T100S100	12	100	100
ND12T80S150	12	80	150	FD12T80S150	12	80	150
ND12T100S150	12	100	150	FD12T100S150	12	100	150

**Table 8 materials-14-05185-t008:** Ultimate bearing capacity and failure mode of specimens.

Specimens	Ultimate Loading Capability (kN)	Failure Mode
ND8T80S100	52	Bottom of bolts fracture
ND8T100S100	62	Bottom of bolts fracture
ND8T80S150	70	Bolts fail in the concrete core and bolts detachment from the bottom plate and the concrete core fractured
ND8T100S150	77	Bottom of bolts fracture
ND10T80S100	85	Concrete core fracture and bolts failure
ND10T100S100	94	Concrete core fracture and bolts failure
ND10T80S150	79	Concrete core fracture and bolts failure
ND10T100S150	85	Concrete core fracture and bolts failure
ND12T80S100	115	Concrete core fracture
ND12T100S100	156	Concrete core fracture
ND12T80S150	151	Concrete core fracture and bolts failure
ND12T100S150	158	Concrete core fracture and bolts failure
FD8T80S100	68	Bolt failed and bottom of bolts crushing
FD8T100S100	81	Bolt failed and bottom of bolts crushing
FD8T80S150	82	Bolt failed and bottom of bolts crushing
FD8T100S150	94	Bolt failed and bottom of bolts crushing
FD10T80S100	70	Bottom of bolts fracture
FD10T100S100	99	Concrete core crushing and bolts failure
FD10T80S150	92	Concrete core crushing and bolts failure
FD10T100S150	99	Concrete core crushing and bolts failure
FD12T80S100	168	Bolt failed and the concrete core crushed
FD12T100S100	204	Bolt failed and the concrete core crushed
FD12T80S150	175	Bolt failed and the concrete core crushed
FD12T100S150	180	Bolt failed and the concrete core crushed

**Table 9 materials-14-05185-t009:** Ductility values of the laboratory samples.

Specimens	$\Delta_{u}\;(\mathrm{mm})$	$\Delta_{y}\;(\mathrm{mm})$	Ductility (i)	Specimens	$\Delta_{u}\;(\mathrm{mm})$	$\Delta_{y}\;(\mathrm{mm})$	Ductility (i)
ND8T80S100	0.84	0.60	1.40	FD8T80S100	2.89	0.32	9.03
ND8T100S100	0.85	0.34	2.50	FD8T100S100	4.76	0.95	5.00
ND8T80S150	1.70	0.32	5.29	FD8T80S150	4.51	0.60	7.51
ND8T100S150	1.99	0.36	5.38	FD8T100S150	3.51	0.33	10.51
ND10T80S100	4.64	0.65	7.13	FD10T80S100	10.00	0.94	10.63
ND10T100S100	5.32	1.57	3.37	FD10T100S100	5.54	0.86	6.44
ND10T80S150	8.78	1.99	4.44	FD10T80S150	8.04	1.64	4.89
ND10T100S150	5.35	1.34	3.93	FD10T100S150	11.10	1.15	9.65
ND12T80S100	5.52	0.70	9.88	FD12T80S100	11.40	0.91	12.52
ND12T100S100	5.01	1.12	10.00	FD12T100S100	19.40	1.96	9.89
ND12T80S150	9.24	1.81	5.71	FD12T80S150	6.04	1.03	5.86
ND12T100S150	5.83	1.75	3.27	FD12T100S150	4.28	0.82	5.22

**Table 10 materials-14-05185-t010:** Characteristics of specimens proposed by Liew et al. [67] and Yousefi and Ghalehnovi [55].

	Specimens	Failure Modes of These Samples	$f_{c}$ $\left(MPa\right)$	Thickness (mm)	Plate Thickness (mm)	Yield Strength of Plates (MPa)	Yield Strength of the Connectors (MPa)	Diameter of the Connectors (mm)
J-hook connectors [67]	N5	Concrete embedment failure	47.7	60	6	310	310	11.7
	N7	Concrete embedment failure	47.7	75	6	310	310	11.7
	N9	Concrete herringbone shear crack	47.7	100	6	310	310	11.7
	HN1	Concrete wedge splitting	43.5	75	6	310	315	15.6
	HN2	Left-strip shear fracture and concrete herringbone shear crack	43.5	75	6	310	315	15.6
	HN3	Concrete embedment failure	43.5	40	6	310	315	15.6
	HN4	Left-strip shear fracture and concrete herringbone shear crack	43.5	75	6	310	340	19.5
	HN9	Concrete wedge splitting	43.5	75	6	310	340	19.5
	HN11	Concrete wedge splitting	43.5	100	6	310	340	19.5
Welded end connectors [56]	6D-1	Connectors shear fracture	27.9	100	6	315	380	20
	8D-2	Left-strip shear fracture and concrete herringbone shear crack	27.9	100	8	315	380	20
	10D-3	Top branch bent down and bottom branch straighten of right-strip and concrete wedge shear	27.9	100	10	315	495	20
	12D-4	Left-strip shear fracture and concrete wedge shear	27.9	100	12	315	516	20
	6Db10-5	Left-strip shear fracture	27.9	100	6	315	380	10
	6Db70-6	Concrete crushing and plate buckling	27.9	100	6	315	495	70
	6Db140-7	Concrete wedge splitting	27.9	127	6	315	516	140
	6Db200-8	Concrete wedge splitting	27.9	186	6	315	615	200

**Table 11 materials-14-05185-t011:** Comparison of the influence of stud-bolts connectors with J-hook and welded end connectors on the maximum shear strength.

Connector	Specimens	Ultimate Strength (kN)	Connector	Specimens	Ultimate Strength (kN)
J-hook connectors [67]	N5	102.0	Stud-bolt	ND10T80S100	78
	N7	94.8		ND10T100S100	94
	N9	93.8		ND10T80S150	79
	HN1	90.5		ND10T100S150	85
	HN2	86.8		ND12T80S100	115
	HN3	84.1		ND12T100S100	156
	HN4	83.7		ND12T80S150	151
	HN9	115.0		ND12T100S150	158
	HN11	120.0		FD8T80S100	68
Welded end connectors [55]	6D-1	71.6		FD8T100S100	81
	8D-2	82.6		FD8T80S150	82
	10D-3	85.2		FD8T100S150	94
	12D-4	86.9		FD10T80S100	70
	6Db10-5	43.6		FD10T100S100	99
	6Db70-6	99.4		FD10T80S150	92
	6Db140-7	114.0		FD10T100S150	99
	6Db200-8	148.0		FD12T80S100	168
Stud-bolt	ND8T80S100	52		FD12T100S100	204
	ND8T100S100	62		FD12T80S150	175
	ND8T80S150	70		FD12T100S150	180
	ND8T100S150	77			

**Table 12 materials-14-05185-t012:** Previous models to predict the shear strength of SCS sandwich panels with different types of connectors.

Study	Model
Ollgaard et al. [31]	$\frac{P}{P_{u}}={\left(1-e^{-18\delta }\right)}^{0.4}$
Cederwall [68]	$\frac{P}{P_{u}}=\frac{2.24\left(\delta -0.058\right)}{1+1.98\left(\delta -0.058\right)}$
Lorence and Kuica [69]	$\frac{P}{P_{u}}={\left(1-e^{0.55\delta }\right)}^{0.3}$
Gattesco and Giuriani [32]	$\frac{P}{P_{u}}=\alpha \sqrt{1-e^{-\beta \delta /\alpha }}+\gamma \delta$
Xue et al. [70]	$\frac{P}{P_{u}}=\frac{\delta }{0.5+0.97\delta }$

## Data Availability

The raw/processed data required to reproduce these findings cannot be shared at this time as the data also form part of an ongoing study.

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
