# Peer review of "Experimental Investigation on the Shear Behaviour of Stud-Bolt Connectors of Steel-Concrete-Steel Fibre-Reinforced Recycled Aggregates Sandwich Panels"

_materials, 2021, doi:10.3390/ma14185185_

Round 1
Reviewer 1 Report
The paper has NOT an adequate scope for Materials journal, considering the aims and scope of the Journal.
The authors present an experimental investigation on the shear behaviour of stud-bolt connectors of steel-concrete-steel recycled aggregates sandwich panels. In the reviewer's opinion, this is a relevant work, which provides interesting findings, but they should be shared with structural and not with the materials scientific community. In contrast to the extensive scientific efforts, the presentation of the manuscript does not comply with the standards of a publication such as Materials. Therefore, the reviewer suggests the authors to prepare a corrected version by carrying out an extensive edition based on the recommendations provided below. The following suggestions and comments should be taken into account before accepting the article for publication.
1. This particular study could be an incremental addition to the knowledge base but it fails to enter into new territory regarding materials science perspective.
Structure-property relationships should be more in depth analysed and presented and discussed. This is a Materials journal, so such property-structure relationships are crucial from scientific point of view. The effect of materials structure, and recycling of material is not considered from materials science point of view.
2. Another critical concern is that the authors have merely reported the observations. Further discussion on the result be included and in-depth analysis be made. Prospects, challenges, future work, limitations, etc. must be discussed in this section.
3. Please also add future research steps which will follow this work.
Author Response
RESPONSES TO Reviewer #1’s COMMENTS:
The authors would like to express their great appreciation towards you for your thorough and detailed review of our manuscript. Without a doubt, the presented ideas and the additional recommended actions have strengthened the article. Your comments were taken into account in the revision of the paper. The places in the text where these suggestions are considered are marked by RED. The answers to all of your valuable comments are given in the following lines.
- COMMENT:This particular study could be an incremental addition to the knowledge base but it fails to enter into new territory regarding materials science perspective.
RESPONSE: From the materials science perspective, it should be mentioned that this paper covers the incorporation of waste materials to manufacture sandwich panels that recently gained high attention worldwide. All in all, to address this comment, more discussion has been added about the importance of using waste materials.
- COMMENT: Structure-property relationships should be more in-depth analysed and presented and discussed. This is a Materials journal, so such property-structure relationships are crucial from the scientific point of view. The effect of materials structure and recycling of material is not considered from the materials science point of view.
RESPONSE: More discussion was added about the structure-property relationships
- COMMENT:Another critical concern is that the authors have merely reported the observations. Further discussion on the result be included and an in-depth analysis is made. Prospects, challenges, future work, limitations, etc. must be discussed in this section.
RESPONSE: This comment was taken into account and more discussion was carried out on the presented results, challenges, future work, limitations, etc.
- COMMENT:Please also add future research steps which will follow this work.
RESPONSE: Future research steps are proposed after the conclusion section.
Again, we appreciate all of your insightful comments. We worked hard to be responsive to them. Thank you for taking the time and energy to help us improve the paper.

Reviewer 2 Report
In the Reviewer opinion the research paper entitled “Experimental investigation on the shear behaviour of stud-bolt connectors of steel-concrete-steel recycled aggregates sandwich panels” is good.
The article investigated a new formula to determine the shear strength of SCS panels with this kind of connectors. According to this study, increasing the diameter of the stud-bolts or using SFC in sandwich panels improve their shear strength and ductility ratio.
Some comments which greatly enhance the understanding of the paper and its value are presented below. Specific issues that require further consideration are:
- The title of the manuscript is matched to its content.
- In the Reviewer’s opinion, the current state of knowledge relating to the manuscript topic has been presented, but the author's contribution and novelty are not enough emphasized.
- In the Reviewer’s opinion, the bibliography, comprising 49 references, is rather representative.
- An analysis of the manuscript content and the References shows that the manuscript under review constitutes a summary of the Author(s) achievements in the field. However, the introduction needs more attention.
- Please improve the quality of drawings and pictures.
- In the Reviewer’s opinion the manuscript should be published in the journal after minor revision.
Author Response
RESPONSES TO Reviewer #2’s COMMENTS:
The authors would like to express their great appreciation towards you for your thorough and detailed review of our manuscript. Without a doubt, the presented ideas and the additional recommended actions have strengthened the article. Almost all of your comments have been included in the paper. The places in the text where these suggestions are considered are marked by GREEN. The answers to all of your comments are given in the following lines.
- COMMENT: In the Reviewer’s opinion, the current state of knowledge relating to the manuscript topic has been presented, but the author's contribution and novelty are not enough emphasized.
RESPONSE: The novelty of this study was discussed in more detail.
- COMMENT: In the Reviewer’s opinion, the bibliography, comprising 49 references, is rather representative
RESPONSE: Thank you for the comment.
- COMMENT:An analysis of the manuscript content and the References shows that the manuscript under review constitutes a summary of the Author(s) achievements in the field. However, the introduction needs more attention.
RESPONSE: The Introduction section was improved.
- COMMENT:Please improve the quality of drawings and pictures.
RESPONSE: The quality of the drawing and pictures has been improved.
Again, we appreciate all of your insightful comments. We worked hard to be responsive to them. Thank you for taking the time and energy to help us improve the paper.

Round 2
Reviewer 1 Report
Thank you on your hard work implementing the comments. The revised version sufficiently and significantly improved this manuscript. I now recommend it for publication.